# Active Health Monitoring of Thick Composite Structures by Embedded and Surface-Mounted Piezo Diagnostic Layer

**DOI:** 10.3390/s20123410

**Published:** 2020-06-17

**Authors:** Tianyi Feng, Dimitrios Bekas, M. H. Ferri Aliabadi

**Affiliations:** Structural Durability and Health Monitoring, Imperial College London, South Kensington, London SW7 2AZ, UK; bekasdim@gmail.com (D.B.); m.h.aliabadi@imperial.ac.uk (M.H.F.A.)

**Keywords:** structural health monitoring, thick composites, temperature influences, impact damage, laser doppler vibrometer

## Abstract

An effective approach for an embedded piezo diagnostic layer into thick composite material is presented. The effectiveness of the approach is assessed in comparison to the surface-mounted layer. The proposed manufacturing alleviates difficulties associated with trimming edges of composites when embedding wires. The Electro-Mechanical Impedance technique is used to access the integrity of the piezoelectric sensors bonding process. Comparisons of ultrasonic guided waves are made between embedded and surface-mounted diagnostic layers and their penetration through and across the thickness of the composites. Temperature influences with the range from −40 °C up to 80 °C on embedded and surface-mounted guided waves are investigated. An investigation is carried out into the relationship between amplitude and time-of-flight with temperature at different excitation frequencies. The temperature has significant but different effects on amplitude and phase-shift of guided waves for the embedded layer compared to the surface-mounted layer. A Laser Doppler Vibrometer is used to identify the blue tack and impact damage. Both embedded and surface-mounted layers are shown to be an effective means of generating detectable wave scatter from damage.

## 1. Introduction

Thick composites as primarily load-carrying structural components have taken an increasingly significant role in aeronautic applications in recent decades [1,2]. They are extensively used, especially in large aircraft structures (Boeing 787 and Airbus 350 XWB), due to their high stiffness and strength, and corrosion resistance. One of the most important safety issues for these large structures is to guarantee the structural integrity and damage tolerance within the limitations of the design [3]. These potential damages, especially impact damage resulting in fibre cracking and delamination, may propagate and ultimately cause failure of critical components. Therefore, it is necessary to continuously monitor the structural integrity of these thick composite structures during their service life. 

A Structural Health Monitoring (SHM) system can provide a real-time assessment of the aircraft’s integrity during operation [1]. It also allows for analysing sensitive components periodically, identifying complex non-visible defects and evaluating the structural integrity through remote sensing [4]. The SHM system based on ultrasonic guided waves (UGW) excited by lead zirconate titanate (PZT) actuators has drawn much attention [5]. UGW can propagate over long distances with less energy loss, and therefore, use of UGW is a potential way to inspect large composite structures [6]. In addition, the use of UGW has been shown to be efficient and effective in identifying and locating damage in composite structures [7]. Therefore, the SHM system can possibly be an effective way to monitor the integrity of thick composites.

PZT transducers are widely used in the SHM system as they are lightweight and relatively inexpensive [8,9,10]. In addition, they can simultaneously exhibit actuator/sensor behaviours, which allows for both passive and active detections [8,9,11]. PZT transducers are usually embedded inside or surface-mounted on the composites. For embedding techniques, Mall and Yocum et al. [12,13,14] proposed two traditional methods: the insertion and cut-out methods. An alternative approach was reported in References [15,16,17,18], where PZT transducers were connected on SMART (Stanford Multi-Actuator Receiver Transduction) Layer^TM^ as an extra ply during lay-up. This layer, based on the circuit printing technique, acted as an extra ply during lay-up instead of simply embedding PZT transducers into composite laminates. 

Other embedding techniques relate to fibre Bragg grating (FBG) sensors. Batte et al. [19] proposed a traditional method by simply inserting the FBG sensors into embedded layers during lay-up, and silicon-impregnated thermoplastic braids were used at the egress/ingress region to protect the connecting area between FBG sensors and composites. However, the edge of composites cannot be trimmed using this method, which is not acceptable in industry manufacturing. To solve this issue, Beukema [20] proposed two embedding methods, the first method involved creating a hollow tube during lay-up, and the FBG sensor could be fed through the hollow tube after curing. The other method involved integrating miniature connector Diamond Micro Interface (DIM) into the embedded FBG sensor. This method was reported to have been qualified by aerospace standard and was used by The National Aeronautics and Space Administration (NASA) in the Mars project. In addition, a different method was introduced by cutting the part of composite layers at the edge area to make the FBG sensor exit much easier [21]. Luyckx et al. [22] and Teitelbaum et al. [23] also proposed to embed miniaturized read-out unit wireless transmissions to thoroughly eliminate the entry point.

Due to the importance of SHM for thick composites, many methods and technologies of damage identification have been investigated. Kirikera et al. [24] and Kesavan et al. [25] used an artificial neural network (ANN) to predict the delamination location for thick glass fibres. Katunin et al. [26] proposed a numerical algorithm of crack detection of fractal dimension for thick composite beams. Sohn et al. [27] used a Laser Doppler Vibrometer (LDV) to detect debonding in the composite wing section. Barely Visible Impact Damage (BVID) detection capabilities of the embedded and surface-mounted PZT transducers were compared in the broad frequency range of the excitation by Dziendzikowski et al. [28]. Other approaches related to embedded PZT transducers have been reported in References [29,30,31,32,33,34,35]. There have also been studies focused on embedded fibre Bragg grating (FBG) sensors. For example, Herszberg et al. [36] attempted to model and predict debonding damage by evaluating strain distribution and vibration response for a glass T-joint. Ghoshal et al. [37] conducted a fatigue test and predicted delamination location for thick carbon fibres.

The traditional wire connecting methods to embed and surface-mount PZT transducers can significantly increase the weight of the host structure. Recently, a developed diagnostic layer [38] utilising inkjet-printed technology has been shown to considerably reduce the weight and thickness of integrated layers. The diagnostic layer was shown to be quite effective for thin composites. Furthermore, the layer was shown to pass tests related to extreme environmental and operational conditions. However, as it will be demonstrated in this paper, surface-mounted PZT transducers cannot effectively detect damage in thick composite since they are designed to generate surface waves (Lamb waves). Hence, embedding sensors into thick composites is a challenge which needs to be overcome for the thick composites.

This paper presents for the first time an investigation into embedding the diagnostic layer into thick composites. In addition, a novel cut-out embedding technique for fabrication of composites which allowed edge trimming after curing has also been proposed. Comparisons of UGW are made between embedded and surface-mounted diagnostic layers and their penetration through and across the thickness of the composite. Investigation is carried out into the amplitude of the first wave packet with different excitation frequencies, and temperature influences on the thick composites for embedded and surface-mounted signals. Furthermore, an impact damage test is conducted and residual signals between the pristine and damage are presented. In addition, a Laser Dropper Vibrometer (LDV) is used to show the scattering of UGW from impact damage for both embedded and surface-mounted layers.

## 2. Experimental Setup

### 2.1. Diagnostic Layer

The diagnostic layer was prepared following the process reported in Reference [38]. For the conductive circuits, a silver-based ink with a nanoparticle concertation of 30–35 wt % was used. The Dimatix Materials Printer (DMP-2580) was used to print circuits onto a transparent Kapton polyimide film with thickness of 25.4 µm. For the inkjet-printed process, the piezo voltage was set at 20 V and an optimised waveform with a jetting frequency of 5 kHz enabled a satisfactory drop formation. The substrate’s temperature during printing was 55 °C and the drop spacing of 35 μm was selected, resulting in a uniform line formation. To decrease the electrical resistivity of the printed circuits, 5 layers of inks were printed on top of each other while the width of the printed lines was selected at 1.4 mm. Sintering of ink took place in the laboratory oven (OF-01E/11E/21E) for 1 h at 135 °C to fuse the conductive particles into a cohesive conductive trace.

### 2.2. Sensor Installation

DuraAct PZT transducers (P-876K025) were used in this experiment. Two-part silver conductive epoxy adhesive resin/hardener (RS 186-3616) was used as a conductive agency to connect the transducers and circuits. For embedding, two transducers were pre-bonded with the Kapton^®^ by adhesive films to prevent delamination between transducers and the Kapton^®^. Figure 1 shows the schematic of the pre-bonding procedure. Two small pieces of adhesive film layers were placed over the Kapton^®^ in the designated position, followed by a PZT transducer. The transducer was then fixed by the blue tape. The Kapton^®^ was placed into an oven for 20 min at 80 °C to cure the conductive agency first to prevent them from flowing and causing short-circuiting during the pre-bonding procedure. Then, the Kapton^®^ was put into a heated vacuum table (G-Sub-1310) to cure for one hour at 150 °C and was cooled down to room temperature with the vacuum to make sure the melting adhesive liquids would not flow out. For surface-mounting, two transducers were connected by epoxy adhesives and cured for 20 min at 80 °C. After pre-bonding, the blue tape was removed carefully.

### 2.3. Manufacturing of Carbon Fibre-Reinforced Composite (CFRP)

The composite specimen was manufactured using Hexply^®^ IM7/8852 unidirectional carbon pre-pregs with a quasi-isotropic stacking sequence of (0°/+45°/−45°/+90°)_9s_. The size of the specimen was 190 × 80 mm, and the average thickness was 8.7 mm. The distance between the two PZT transducers was 110 mm. The distance from the PZT transducer to each edge was 40 mm to eliminate reflection effects on signals. The embedding position was in the middle of the panel between two +90° plies to have the minimum effects on integral strength under the uniaxial loading condition.

In this paper, a novel cut-out embedding and manufacturing method is proposed. A key difficulty with previous PZT embedding methods is the leading wires coming out of the edge of the composites. This will prevent trimming of the edge of composites, which is a requirement for industrial manufacturing and assembly. There is also a potential risk that leading wires suspended in the air may be broken during transferring and testing. In this method, a small edge area is cut out from the upper pre-pregs of the specimen to trim the edge of the specimen and keep the embedded leading wires after curing. The pre-pregs ply-cutting plan is divided into three parts, the bottom pre-pregs, the main upper pre-pregs and the cut-out pre-pregs, as is shown in Figure 2a, and a schematic of the assembly is shown in Figure 2b.

Figure 3 illustrates the schematic of the cross-sectional area of the cut-out part of the specimen. During the initial stage of the embedding process, a small area of release film was placed over the upper cut-out area to easily separate the embedded Kapton^®^ from the bottom pre-pregs at that area. A layer of resin film (Hexply^®^ M56) was then placed over the bottom pre-pregs to increase the bonding properties between the bottom pre-pregs and the Kapton^®^. The embedded Kapton^®^ was then laid in the designated position of the middle layer of the panel. A layer of resin film was placed over the embedded film to prevent delamination, followed by a blank Kapton^®^ to prevent circuits from short-circuiting during curing. Another layer of the resin film was then placed over the blank Kapton^®^ to increase the bonding property between the main upper pre-pregs and the Kapton^®^.

For the remainder of the embedding process, the main upper pre-pregs were then placed on top of the resin film, and the exposed Kapton^®^ at the cut-out area was attached tightly to the vertical side of the main upper pre-pregs. Subsequently, the release film at the cut-out area was removed and the upper cut-out pre-pregs were then inserted into the cut-out area. Once it had been inserted, the exposed part of the Kapton^®^ was placed over the upper cut-out area of the specimen. When the lay-up was finished, a small piece of release film was placed over the exposed circuits of the Kapton^®^ to protect the circuits from damage. Finally, the edge of the release film was sealed using the blue tape to prevent the epoxy in the pre-pregs from flowing onto the circuit area during the curing procedure.

At the end of the manufacturing process, the composite specimen was inspected via C-Scan (DolphiCam CF08), and non-destructive test (NDT) results are shown in Figure 4. As can be seen in Figure 4a, the position of embedded PZT transducer can be detected, which is in the middle of the panel according to the vertical/horizontal B-Scan. In Figure 4b, the non-embedding areas show that the bonding quality of the specimen is good and there is no defect in the specimen.

For the surface-mounting procedure, two small layers of adhesive film were placed over the mould side of the specimen for bonding the Kapton^®^ and the specimen. The surface-mounted Kapton^®^ was placed over the adhesive films at the designated position, where two PZT transducers were in same positions as these embedded transducers. Then, these extra adhesive films were removed slightly using a knife, and the surface-mounted Kapton^®^ was fixed using blue tape. The panel was placed on the heated vacuum table to cure for one hour at 150 °C and allowed to cool down to room temperature to make sure the melting adhesive liquids would not flow out. After bonding, the specimen was trimmed, and two connectors (RS 514-4408, operating temperature range: −40 °C to 85 °C) were placed to designated areas to connect embedded and surface-mounted circuits and bonded by the super glue (RS 473-445, operating temperature range: −50 °C to 80 °C). The final specimen with a surface-mounted diagnostic layer is shown in Figure 5. 

## 3. Sensor Integrity

### 3.1. Literature Review of Electrical-Mechanical Impedance Technique

The Electrical-Mechanical Impedance (EMI) method based on active sensing (normally embedded or surface-mounted PZT transducers into/on the host structure) is one of the most promising methods in SHM technology. The theory of the EMI method is based on the piezoelectric effect that a PZT transducer can generate strain conversed by applying a harmonic voltage [39]. This method can not only evaluate local damage severities but can also detect bonding conditions between the PZT transducer and host structures. Sharif-Khodaei et al. [39] experimentally and numerically evaluated BVID and debonding on a composite panel, and further numerically applied the EMI technique to a complicated stiffened composite panel. The debonding and BIVD were detected when root-mean-square deviation (RMSD) values of the impedance were higher than normal ones. In addition, Zou et al. [40] used the EMI technique to numerically study the debonding detection of aluminium substrate based on the boundary element method and also experimentally validated it with numerical results. The damage indexes (DI): RMSD, mean absolute percentage deviation (MAPD) and correlation coefficient deviation (CCD), were used to evaluate the debonding in this paper. It has been found that the frequency range of 70–85 kHz could be used for detecting the damage using MAPD and CCD values.

On the other hand, the EMI technique based on embedded and surface-mounted PZT transducers was widely used in civil engineering. Providakis et al. [41] used the EMI technique to numerically study the debonding detection based on DI—room mean square error (RMSE)—for a reinforced concrete (RC) beam. In addition, Chalioris et al. [42] numerically studied the frequency range for detecting cracks and yielding damage by using RMSD value for RC beams with surface-mounted PZT transducers. The frequencies for detecting cracks were 80 kHz and 180 kHz, and for detecting yielding damage were 60 kHz and 160 kHz. Further, Voutetaki et al. [43] proposed a novel, portable and real-time Wireless Impedance/Admittance Monitoring System (WiAMS) to experimentally access failures in shear on RC beams. The frequency range was 10 to 260 kHz, and the DI values (RMSD) successfully detected the damage for both embedded and surface-mounted PZT transducers. It was reported that the RMSD value of final shear failure for the PZT mounted to critical span was remarkably higher than that of other shear spans of the beam. Furthermore, the WiAMS was experimentally studied for detecting cracks and shear diagonal cracking of the RC column with embedded and surface-mounted PZT transducers based on RMSD values by References [44] and [45], respectively.

In addition, the EMI technique can also be used for accessing transducers’ fractures, degradation of mechanical/electrical properties of transducers and the integrity of the bonding properties between the PZT transducer and its host structure [46,47]. Previous research had found that any change in host structure would change its vibration behaviour and indirectly affect the admittance properties of a PZT transducer [48,49]. The admittance change in imaginary parts will determine the integrity structure between the transducer and its host structure and that change in real parts will determine the damage distance from the PZT transducer [39]. Park et al. [47] accessed the integrity of bonding properties by comparing the slope of imaginary parts of the admittance for surface-mounted PZT transducers by using different bonding materials. Voutetaki et al. [43] pointed out that different bonding situations between embedded and surface-mounted PZT transducers would cause shifts in the slope of imaginary parts of admittance.

### 3.2. Theory of the EMI Technique Based on Admittance Measurements

The piezoelectric effect of a PZT transducer can be expressed by [46]:(1)$$S_{i}=s_{ij}^{E}T_{j}+d_{mi}E_{m}$$
(2)$$D_{m}=d_{mi}T_{i}+\varepsilon_{mk}^{T}E_{k}\;$$
where *S* denotes the mechanical strain, *s* denotes the mechanical compliance, *T* denotes the mechanical stress, *d* denotes the piezoelectric coupling constant, *E* denotes the electric field, *D* denotes charge density, ε is the dielectric constant of the PZT transducer, subscripts *i* and *j* are directions of the stress and subscripts *m* and *k* are directions of strain and electric field. Both electric filed *E* and mechanical stress *T* are measured at zero electric field and zero stress, respectively.

For a one-dimensional situation, the total electric current, *I*, flowing through the PZT transducer can be expressed when *T* = 0,
(3)$$I=i\omega \iint D_{3}dxdy=i\omega \iint \varepsilon_{33}^{T}\left(1-\delta \right)E_{3}dxdy$$
where ω is the angular frequency, δ is the dielectric loss tangent of the PZT transducer and *x* and *y* denote the coordinates of the PZT transducer.

The formula of electrical admittance for a free-free boundary condition PZT transducer *Y_free_*(ω) is given in Reference [47] as:(4)$$Y_{free}(\omega )\;=\;\frac{I}{V}\;=\;i\;\omega \frac{wl}{t_{c}}\left(\varepsilon_{33}^{T}\left(1-i\delta \right)\right)$$
where *V* denotes the applied voltage, and *w*, *l* and *t_c_* denote the width, length and thickness of the PZT transducer, respectively.

For a surface-mounted PZT transducer, the formula of the electrical admittance *Y_surface_*(ω) is shown in Reference [47] as:(5)$$Y_{surface}(\omega )\;=\;i\;\omega \frac{wl}{t_{c}}\left(\varepsilon_{33}^{T}\left(1-i\delta \right)-d_{31}^{2}{\hat{Y}}_{P}^{E}+\frac{Z_{a}\left(\omega \right)}{Z_{a}\left(\omega \right)+Z_{s}\left(\omega \right)}\right)d_{31}^{2}{\hat{Y}}_{P}^{E}\;\left(\frac{\tan kl}{kl})\right)$$
where *d_31_* is piezoelectric strain constant, ${\hat{Y}}_{P}^{E}$ is the complex Young’s modulus of the PZT at zero electric field, $Z_{a}\left(\omega \right)$ and $Z_{s}\left(\omega \right)$ are the mechanical impedance of the host structure and that of PZT transducer respectively, and *k* is the wavenumber of the PZT transducer, which is defined as:(6)$$k=\omega \sqrt{\frac{\rho }{{\hat{Y}}_{P}^{E}}}\;$$
where ρ denotes the mass density of the PZT transducer.

Mathematical expressions for an embedded PZT transducer depend largely on the manufacturing conditions of embedding. A three-dimensional (3D) model of the electrical admittance *Y_embedded_*(ω) suitable for embedding PZT into concrete was proposed in Reference [48].

### 3.3. Bonding Properties

In this experiment, a SinePhase Impedance Analyser (Model 16777K) was used to measure the imaginary and real parts of the admittance of PZT transducers to compare with that of a free-free transducer. The frequency range was from 0 to 1000 kHz. Figure 6 depicts the imaginary and real parts of the admittance for these free-free, embedded and surface-mounted PZT transducers with a range of excitation frequencies, respectively. In Figure 6a, the first resonance vibration features for free-free, embedded and surface-mounted PZT transducers appear at 193 kHz, 319 kHz and 240 kHz, respectively. In addition, both slopes of the imaginary parts of the admittance for the embedded and surface-mounted PZT transducers are higher than that of the free-free transducer, from 0 to 20 kHz. Furthermore, those of embedded PZT transducers are much higher those of surface-mounted PZT transducers, and this is because boundary conditions between embedded and surface-mounted PZT transducers are different. These embedded transducers were bonded from both sides, whilst these surface-mounted PZT transducers were only bonded from one side. In Figure 6b, the first resonance vibration features for free-free, embedded and surface-mounted PZT transducers appear at 195 kHz, 337 kHz and 274 kHz, respectively. According to the EMI technique, there is no obvious slope shifting of the imaginary parts of admittance between two embedded and surface-mounted PZT transducers at a low frequency range of 0–20 kHz. Therefore, both embedded and surface-mounted PZT transducers have ideal bonding properties with the host structure.

## 4. Ultrasonic Guided Waves

A National Instruments (NI) PXIe-1073 was used for signal measurements, where a NI PXI-5412 arbitrary signal generator was used for signal generation, and a NI PXI-5105 digitiser was used to record Lamb wave signals. The specimen was measured at excitation central frequencies of 50 kHz, 100 kHz, 150 kHz, 200 kHz and 250 kHz under room temperature. A five-cycle Hanning-windowed toneburst signal was used, and the amplitude and sampling frequency were 6 V and 100 MHz, respectively. The reversibility and repeatability of two paths for both embedded and surface-mounted signals were found to be good. To summarise the relationship between the peak amplitude of the first wave packet at five excitation frequencies, Hilbert transform was used to obtain envelope signals, then the amplitude was selected from the first peak value of each envelope signals. As is shown in Figure 7, the amplitude of the surface-mounted signal is larger than that of the embedded signal, except for at 100 kHz. This is due to tuning properties, where S_0_/A_0_ mode exhibits various amplitudes under different frequencies [50,51].

## 5. Influence of Temperature on Ultrosonic Guided Waves

The specimen was placed in a thermal vacuum chamber J2235 (TAS Ltd.) to investigate temperature influences on embedded and surface-mounted signals. The temperature ranges were from –40 up to 80 °C (limited to the operation temperature of connectors). Five-cycle Hanning-windowed toneburst signals at 50 kHz, 100 kHz, 150 kHz, 200 kHz and 250 kHz were used during operation. Figure 8 and Figure 9 show relationships between amplitude and time-of-flight with temperature. The amplitude was measured from the peak amplitude of the first wave packet under different temperatures. The value of time-of-flight was measured at the time at the peak amplitude of the first wave packet.

As can be seen in Figure 8a, when increasing the temperature, the amplitude at 50 kHz, 100 kHz, 200 kHz and 250 kHz increases, while the amplitude at 150 kHz decreases for embedded signals. In Figure 8b, the amplitude increases at 150 kHz and 200 kHz and decreases at 50 kHz and 100 kHz when increasing the temperature for surface-mounted signals. In addition, the amplitude increases from −40 to 40 °C and then decreases from 40 to 80 °C at 250 kHz when increasing the temperature. As can be seen in Figure 9, the value of time-of-flight increases when increasing the temperature for both embedded and surface-mounted signals.

Therefore, the change in temperature can significantly affect the amplitude and phase-shift of UGW at different frequencies for both embedded and surface-mounted signals. The amplitude increases when increasing the temperature at most frequencies. Inversely, this relationship switches at 150 kHz for embedded signals, and at 50 kHz and 100 kHz for surface-mounted signals, respectively. Furthermore, the velocity of UGW decreases with increasing temperature at all five frequencies.

There are five factors which affect the inverse effects of amplitude and phase-shift of travelling times. First, the elastic modulus of composites is a key parameter for thermal variations. The elastic modulus associates with the mechanical stiffness. When increasing the temperature, the elastic modulus decreases significantly as a result of decreasing stiffness, which causes the velocity reduction in guided waves [52,53,54,55,56,57]. Second, the piezoelectric properties of PZT transducers (*d*_31_ and *g*_31_) vary significantly with temperature. The piezoelectric constants *d*_31_ and *g*_31_ associate with shear strain and piezo sensitivity, respectively. The dielectric constant also has a linear relationship with temperature. Both these constraints depend on the output voltage (amplitude) of signals [54,56,57].

In addition, thermal expansion causes changes in panel thickness, piezo dimensions, propagation distance of guided waves and material density, and these key factors are used to compute dispersion curves of UGW in different temperatures [52,54]. The various amplitude of UGW at different frequencies under various temperatures can be affected by wave dispersions as the frequency and temperature depend on material damping [55]. Phase velocities of shear and longitudinal waves are also dependant on wave dispersion [52].

Tuning properties of UGW are also affected by dispersive factors [50,51]. The experimental results of the composites have shown that (1) for embedded signals, A_0_ mode dominates at 50 kHz and S_0_ mode dominates at 100–250 kHz, and (2) for surface-mounted signals, A_0_ mode dominates at 50–100 kHz and S_0_ mode dominates at 150–250 kHz. Therefore, the different exciting A_0_/S_0_ mode in embedded and surface-mounted signals also determines these different relationships between amplitude/time-of-flight with temperature.

Furthermore, thermal expansion and the temperature-induced change also affect PZT transducers and their bonding properties [53]. In this experiment, resin film was used to bond the embedded PZT transducers to meet the specification of carbon fibre pre-pregs fabrication, whilst adhesive film was used to bond surface-mounted PZT transducers. Experimental results and more details related to adhesive films were reported in previous research [58]. Hence, the selection of adhesive film used to bond the PZT transducer is one of factors to affect the experimental results.

## 6. Damage and Scattering

HexPly^®^ IM7/8552 unidirectional carbon pre-pregs were used in this experiment. The lay-up stacking sequence was (0° /+45° /−45° /+90°)_9s_ and the size of the specimen was 300 × 225 × 8.94 mm. DuraAct PZT transducers (P-876K025) integrated with Kapton^®^ films were used for embedding and surface-mounting. The distance between two transducers was 110 mm. A Laser Doppler Vibrometer (LDV) was used to investigate wave scattering from two separate scenarios of a blue tack and impact damage. The LDV model was PSV-500-3D-M from Polytec. Co. in the experiment. An Agilent 33120A arbitrary waveform generator was used to generate chirp signals ranging from 50 to 500 kHz. A burst period was set as 15 ms before the next actuation to attenuate the signal to background level. The signal amplitude voltage was 2 V, and the amplifier (WMA-300, Falco System, maximum ± 150 V) was used to amplify the input voltage to ± 100 V to the PZT actuator. The maximum sampling frequency was 2.56 MHz, with 2048 samples averaged with 200 measurements.

### 6.1. Damage Detection

In order to evaluate the thickness influence on UGW and the distance effect between the damage and sensors on the thick composite laminate, a weighted blue tack was placed on the opposite side of surface-mounted PZT transducers and positioned closer to a PZT transducer (see Figure 10a). As is shown in Figure 10b, the distance from PZT 2 to the edge is 40 mm to eliminate the reflection effects of the wave propagation, and the position of the blue tack is closer to PZT 1 but further from PZT 2. The LDV was used to identify the blue tack position. The spacing between each scan grid was 0.75 mm and the scan area was about 70 × 70 mm, with total of 8649 scan points (93 × 93).

Chirp signals were reconstructed in five-cycle tuneburst signals with Hanning-window at different frequencies using the algorithm based on equations in Reference [59]. Figure 11 shows LDV scanning results for both embedded and surface-mounted PZT transducers at 50 kHz, 100 kHz, 150 kHz, 200 kHz and 250 kHz. In the case of the blue tack, the following observations can be made: 50 kHz, 100 kHz and 150 kHz can interact with this blue tack damage and there is no scattering at 200 kHz and 250 kHz for both embedded and surface-mounted PZT transducers.

From the LDV results, the following conclusions can be drawn: (a) low excitation frequencies are suitable for the type of blue tack damage detection, (b) both embedded and surface-mounted PZT transducers are pronounced at 50 kHz, 100 kHz and 150 kHz and (c) the distance between the PZT transducer and blue tack is not one of the factors that affect the identification in this situation.

### 6.2. Impact Test

An impact test was then carried out to create a real impact damage scenario. HexPly^®^ M21/134/T800S unidirectional carbon pre-pregs were used in this experiment. The lay-up stacking sequence was (0°/+45°/–45°/+90°)_4s_ and the size of the specimen was 300 × 225 × 8.8 mm. The PZT transducers’ positions were the same as the previous ones. The impact position was same as the blue tack position which was on the opposite side of the surface-mounted PZT transducers. An INSTON CEAST 9350 Drop Tower (Figure 12) was used for the impact test. The hemisphere diameter of the impactor was 20 mm and the mass of the impactor was 2.50 kg. The impact energy started from 40 J and increased by 5 J steps every time until the damage was detected. Damage was initiated with impact energy of 60 J (4.9 m/s for the velocity). After the impact, small cracks, with total length 16 mm, were observed on the impact surface, as is shown in Figure 13. Figure 14 shows the NDT result after the impact test, the B-Scan horizontal part shows that delamination is at the second layer of the composite laminate and there are some cracks at the surface. The C-scan (amplitude) part shows that the damage area is about 300 mm^2^.

### 6.3. Signal Comparison

Figure 15 shows comparisons between pristine and damage signals for both embedded and surface-mounted PZT transducers at 50 kHz, 100 kHz, 150 kHz, 200 kHz and 250 kHz. The peak amplitude of the first wave packet for embedded and surface-mounted signals decreased after impact. Figure 16 plots the relationship between residual peak amplitude of the first wave packet (differences between pristine and damage envelope signals) and frequency for embedded and surface-mounted signals. As can be seen in Figure 16, the residual amplitude of surface-mounted signal is higher than that of the embedded signal, at 50–250 kHz. This is due to different positions between embedded and surface-mounted PZT transducers and the impact side of the composite laminate, so the Lamb wave actuated by the surface-mounted PZT transducer has to transfer the whole thickness of the composite panel, whilst the wave actuated by the embedded PZT transducer only transfers half of the thickness of the panel. Hence, surface-mounted signals have higher attenuation than that of embedded signals.

### 6.4. Impact Damage Detection

The impact damage was identified by the LDV. There were total of 11,025 scanning grids arrayed with 105 × 105. The average spacing between each scan grid was 0.7 mm, and the scanning area was about 74 × 74 mm. After post-processing, chirp signals were then reconstructed into five-cycle Hanning-windowed toneburst signals at 50 kHz, 100 kHz, 150 kHz, 200 kHz and 250 kHz. As is shown in Figure 17, the impact damage can be identified by embedded and surface-mounted PZT transducers at 50–250 kHz. In conclusion, (a) all 50–250 kHz were pronounced for the impact damage, (b) both embedded and surface-mounted PZT transducers could identify the impact damage and (c) the distance between the PZT transducers and impact damage did not affect to the damage detection in this situation.

## 7. Conclusions

In this paper, active embedded and surface-mounted piezo diagnosed layers were used to study UGW propagations in thick composites. A novel strategy for SHM of thick composites was investigated by embedding diagnostic layers in the structure. The developed embedding technique alleviated difficulties of the wires coming out of the edge of the composites, which has so far proven to be a key impediment of embedding sensors into composites.

The EMI technique was used to access the integrity of bonding properties between PZT transducers and their host structure. It has been found that the slope of the imaginary parts of the admittance of embedded PZT transducers is higher than that of surface-mounted and free-to-free PZT transducers at 0–20 kHz. In addition, the first resonance vibrations to free-to-free, embedded and surface-mounted PZT transducers are 193 kHz, 319 kHz and 240 kHz for the imaginary parts of the admittance, and are 195 kHz, 337 kHz and 274 kHz for the real parts of the admittance. Furthermore, the bonding integrity of embedded PZT transducers meets the double-side bonded requirement.

The comparison of UGW of embedded and surfaces-mounted layers at 50–250 kHz was also presented. It has been found that the amplitude of the surface-mounted signal is higher than that of the embedded signal, except for at 100 kHz.

Temperature influences on UGW were then studied. The peak amplitude of the first wave packet and time-of-flight were affected by the temperature. When increasing the temperature, the amplitude exhibits positive/negative linear relationships with different excitation frequencies and the time-of-flight will delay.

For LDV scanning: (1) the distance between the PZT transducer and blue tack does not affect the damage detection results, (2) for blue tack damage: 50–150 kHz can identify the damage for embedded and surface-mounted PZT transducers, (3) for impact damage: 50–250 kHz can identify the impact damage for embedded and surface-mounted PZT transducers and (4) surface-mounted signals have higher attenuations than embedded signals after impact.

Further research will focus on damage detection and localisation for thick composites based on mathematical algorithms using active embedded and surface-mounted piezo diagnosed layers.

## Figures and Tables

**Figure 1 sensors-20-03410-f001:**
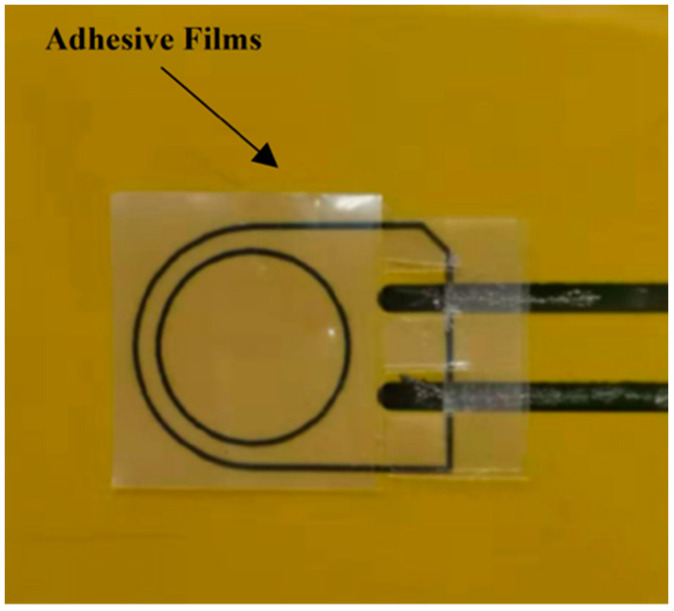
Schematic of the pre-bonding procedure.

**Figure 2 sensors-20-03410-f002:**
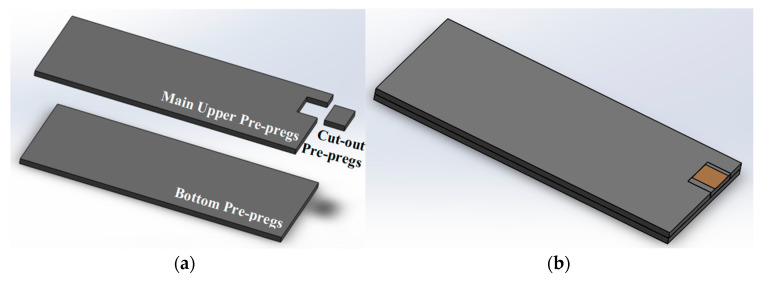
Schematics of (**a**) ply-cutting plan, (**b**) pre-pregs assembly model.

**Figure 3 sensors-20-03410-f003:**
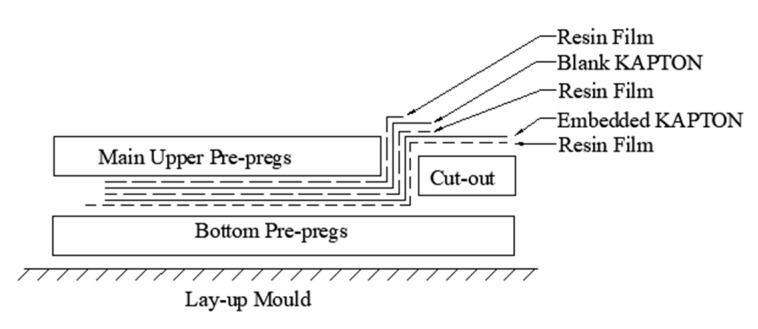
Schematic of the embedding procedure.

**Figure 4 sensors-20-03410-f004:**
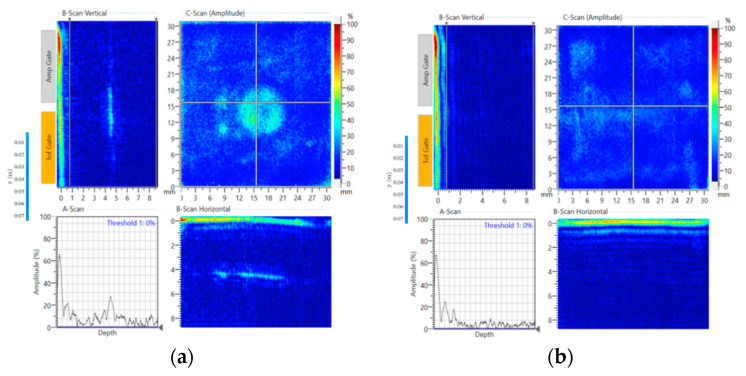
C-Scan results for (**a**) embedded areas, (**b**) non-embedded areas.

**Figure 5 sensors-20-03410-f005:**
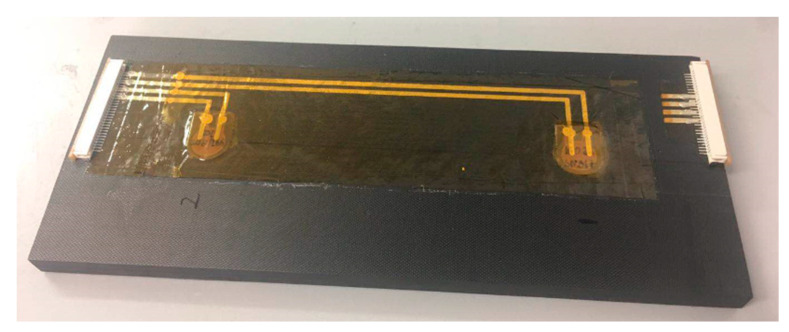
Surface-mounted diagnostic layer.

**Figure 6 sensors-20-03410-f006:**
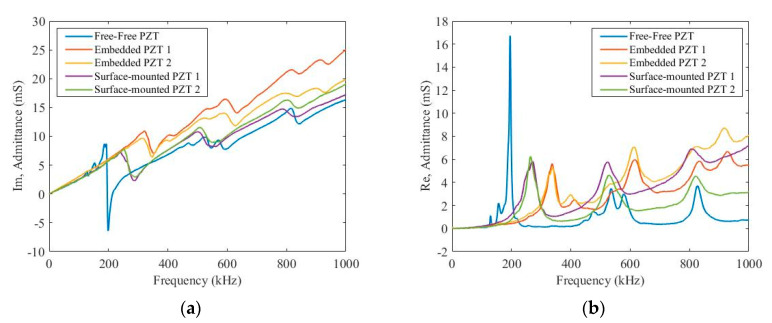
Electrical-Mechanical Impedance (EMI) admittances of (**a**) imaginary parts and (**b**) real parts of free-free, embedded and surface-mounted lead zirconate titanate (PZT) transducers.

**Figure 7 sensors-20-03410-f007:**
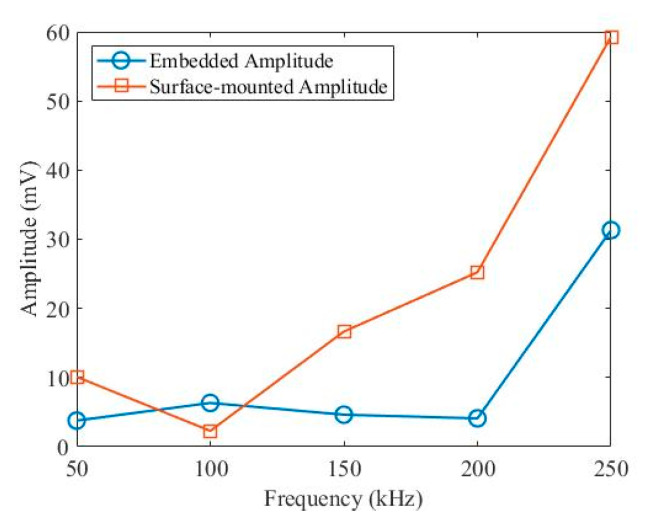
Comparisons of first peak amplitudes between embedded and surface-mounted PZT transducers.

**Figure 8 sensors-20-03410-f008:**
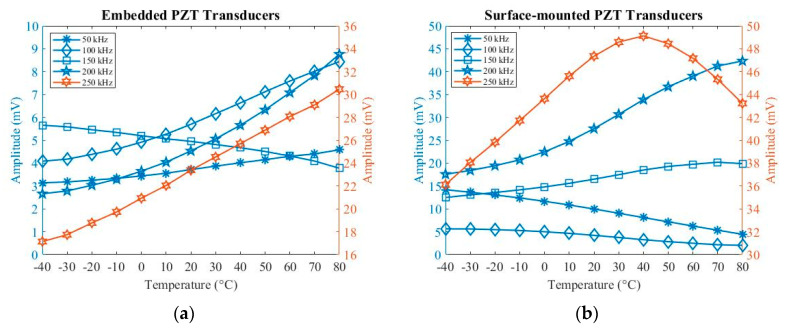
Temperature influences on amplitude for (**a**) embedded and (**b**) surface-mounted signals.

**Figure 9 sensors-20-03410-f009:**
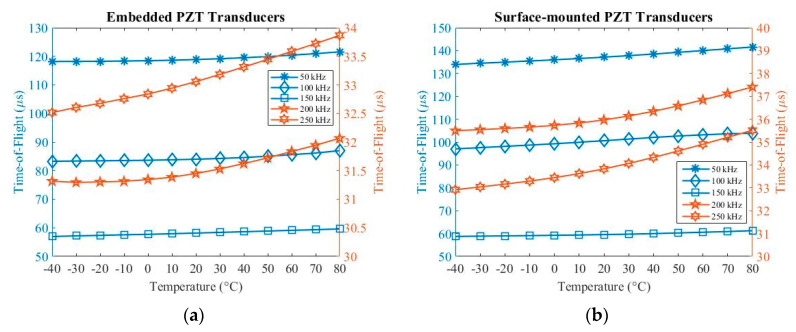
Temperature influences on time-of-flight for (**a**) embedded and (**b**) surface-mounted signals.

**Figure 10 sensors-20-03410-f010:**
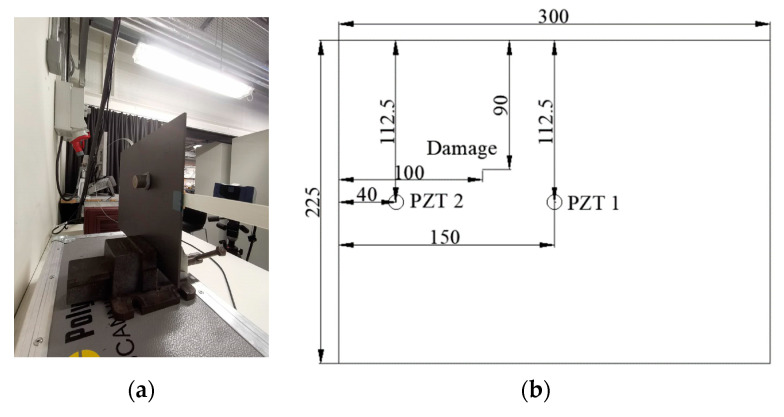
(**a**) Weighted blue tack and (**b**) schematic of the damage position.

**Figure 11 sensors-20-03410-f011:**
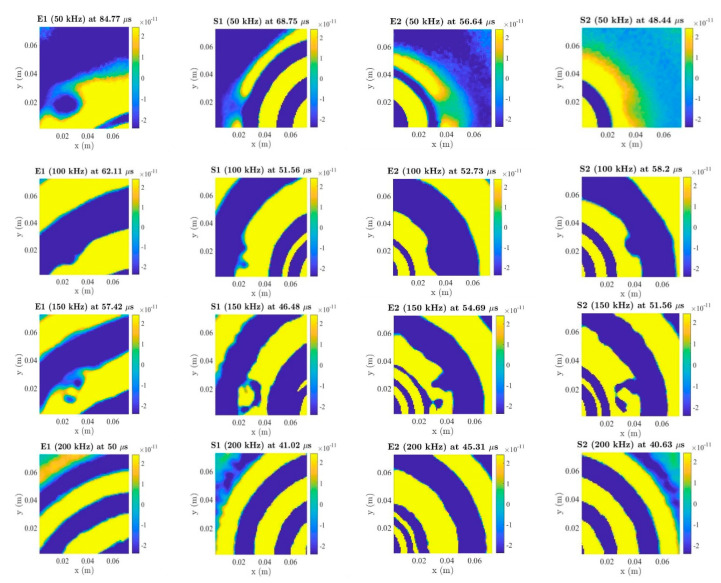
Laser Dropper Vibrometer (LDV) results for the blue-tack (E: embedded, S: surface-mounted).

**Figure 12 sensors-20-03410-f012:**
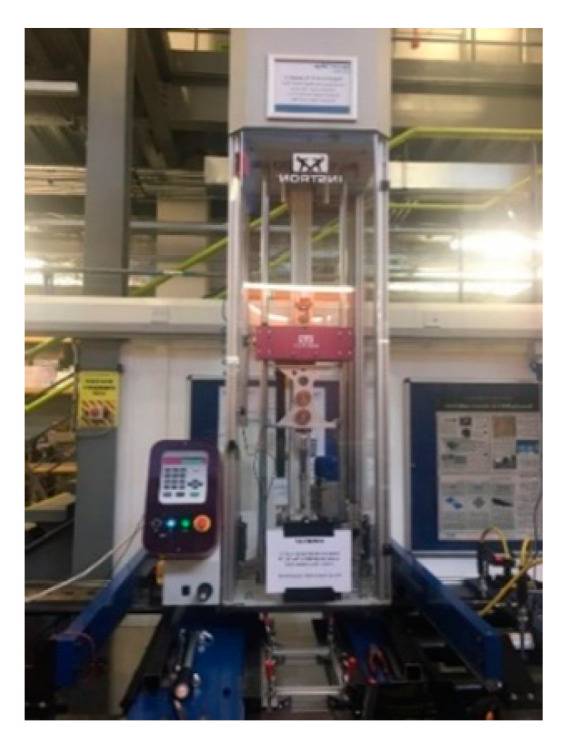
INSTON CEAST 9350 drop tower.

**Figure 13 sensors-20-03410-f013:**
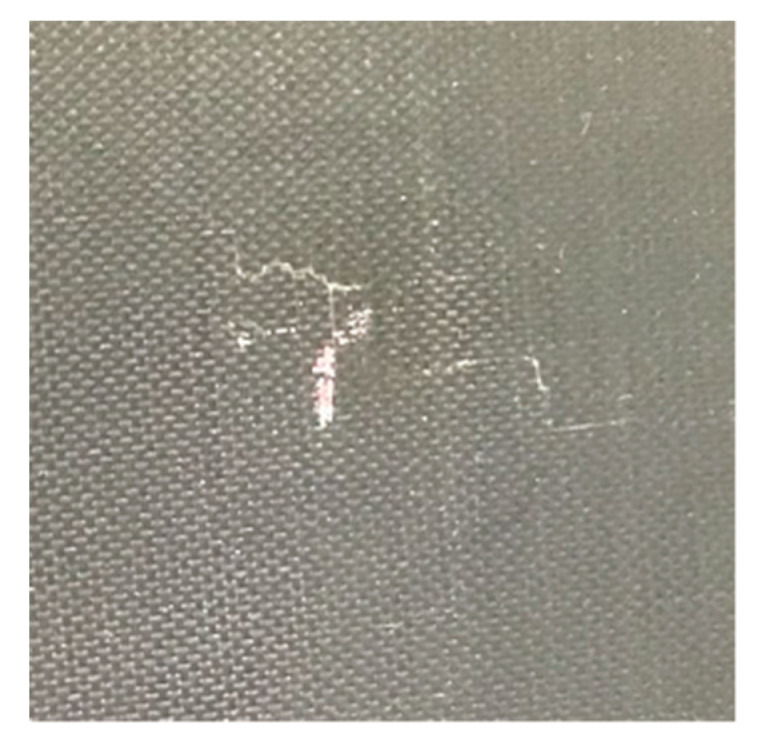
Surface cracks after impact.

**Figure 14 sensors-20-03410-f014:**
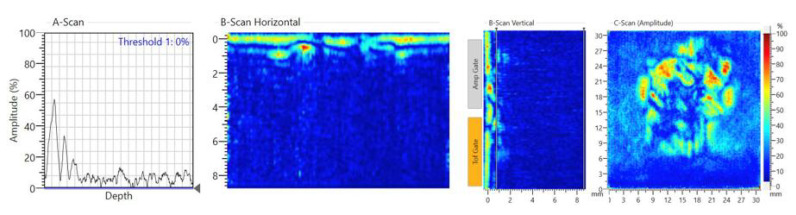
Non-destructive test (NDT) result after impact.

**Figure 15 sensors-20-03410-f015:**
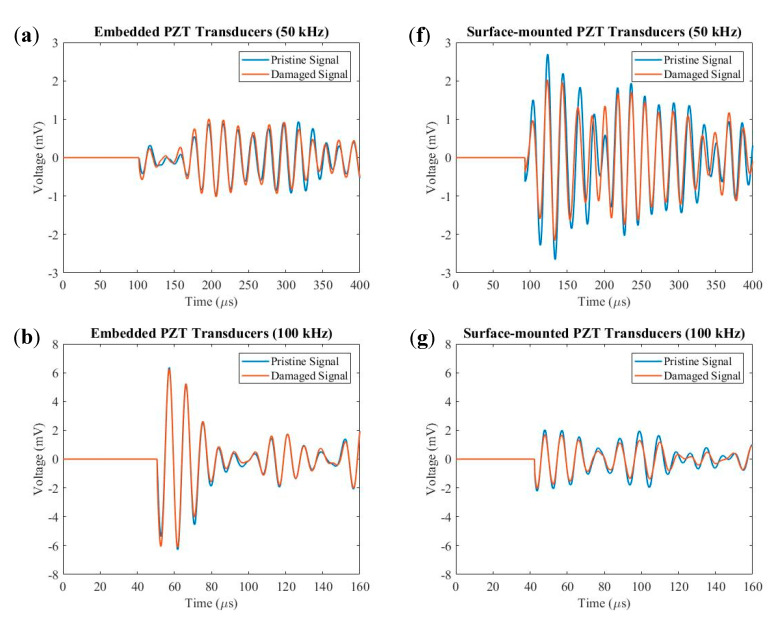
Signal comparisons between pristine and damage signals for (**a**)–(**e**) embedded and (**f**)–(**j**) surface-mounted PZT transducers at 50–250 kHz.

**Figure 16 sensors-20-03410-f016:**
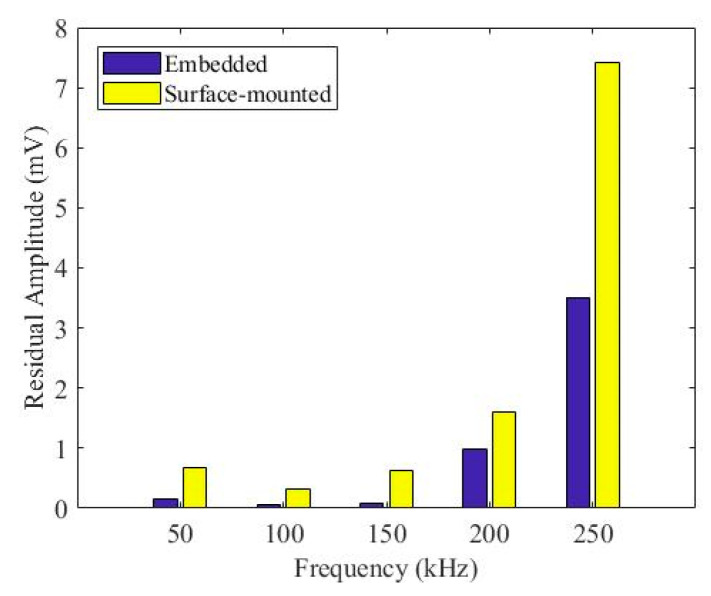
Residual amplitude comparison between embedded and surface-mounted signals.

**Figure 17 sensors-20-03410-f017:**
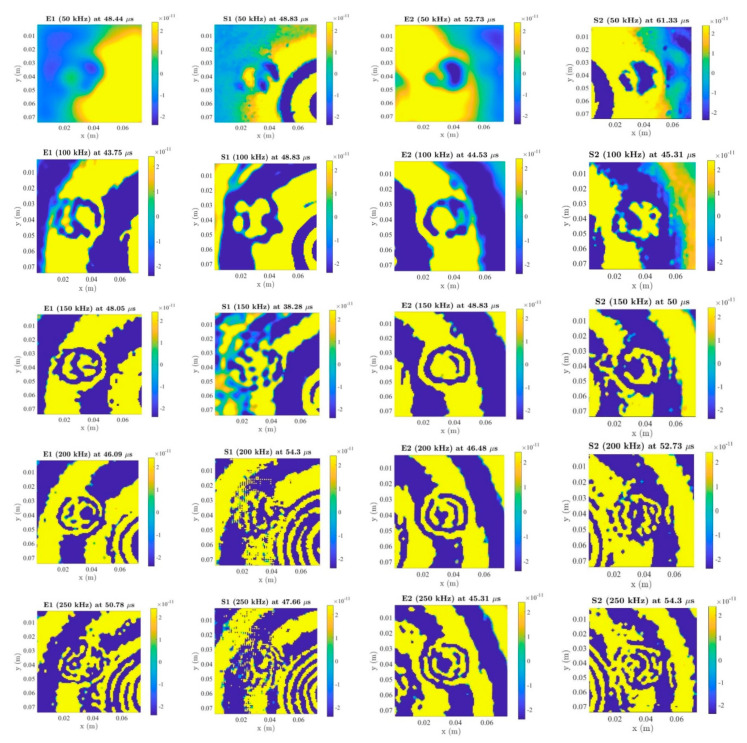
LDV results for impact damage (E: embedded, S: surface-mounted).

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
