# Peer review of "Active Health Monitoring of Thick Composite Structures by Embedded and Surface-Mounted Piezo Diagnostic Layer"

_sensors, 2020, doi:10.3390/s20123410_

Round 1
Reviewer 1 Report
Please see attached file.

Author Response
Dear Sir or Madam,
We are grateful for reviewer’s suggestion and the manuscripts has been revised taking into consideration those comments. The revised manuscript has been attached in below files.
1. In general, the figures are of low quality and resolution. Also, some of these figures lack necessary elaboration in the text (e.g. Fig. 4, Fig. 14, etc). They should be well improved to enhance the readability the manuscript.
- In the revised manuscript, the figure qualities and resolutions have been improved, please see Fig.6, Fig.7, Fig.8, Fig.9 and Fig.15 in the manuscript.
- The explanation of Fig.4 has been added (the red part in section 2.3).
- In Fig. 14, there was an explanation for the NDT result:” Fig.14 shows the NDT result after impact test, the B-Scan horizontal part shows that delamination was at the second layer of the composite laminate and there were some cracks at the surface. The C-scan (amplitude) part shows that the damage area was about 300 mm2.”
2. Introduction - How did the application of the technology on "thick" composites incurred challenges and how were these challenges solved? These should be clearly stated and explained in order to demonstrate the novelty of the manuscript.
- These sentences have been added in the introduction part: “However, as it will be demonstrated in this paper, surface-mounted PZT transducers cannot effectively detect damage in thick composite since there are designed to generate surface waves (Lamb waves). Hence, embedding sensors into thick composites is a challenge which needs to be overcome for the thick composites.”
3. Bonding properties - It is difficult to see from the EMI results why "both embedded and surface-mounted PZT transducers have ideal bonding properties with host structures'". Also, a couple of members of the same group (e.g.Z. Sharif-Khodaei and F. Zou) have done systematic studies on the use of EMI measurements for damage detection and transducer self-diagnosis. Their works could probably offer some useful insights.
- Already provide the literature review of EMI technique in section 3.1. To evaluate the integrity of bonding properties between PZTs and host structure, new information by comparing the difference of two embedded PZTs and two surface-mounted PZTs. Because there is no change in the imaginary part of the admittance, so the integrity is good (In section 3.3).
4. Considering the relatively large thickness of the composite sample used, have the authors checked if ultrasonic waves of low frequencies (e.g. 50 kHz and 100 kHz) still travelled as guided waves? Also, what were the wave modes? This could be one of the reasons behind the relatively inconsistent relationships between amplitude, time-of-flight and temperature.
The author did a basic research at the start which used the LDV to scan the 9 mm thickness composite panel to 1) study the wave propagations and 2) distinguish the wave modes for embedded/surface-mounted signals at 50-250 kHz.
- LDV results have demonstrated that low frequencies (50 kHz and 100 kHz) still travelled as guided waves in 9 mm thickness composites.
- Below table is previous LDV scanning results of S0/A0 mode distinctions. Overall, 1) for embedded signals, A0 mode dominates at 50 kHz, and S0 mode dominates at 100-250 kHz; 2) for surface-mounted signals, A0 mode dominates at 50-100 kHz, S0 mode dominates at 150-250 kHz.
|
Frequency |
Embedded Signals |
Surface-mounted Signals |
|
50 kHz |
S0 (weak, can be ignored) & A0 |
A0 |
|
100 kHz |
S0 & A0 |
S0 (weak, can be ignored) & A0 |
|
150 kHz |
S0 & A0 |
S0 & A0 |
|
200 kHz |
S0 & A0 |
S0 & A0 |
|
250 kHz |
S0 & A0 |
S0 & A0 |
- The theory of A0/S0 amplitudes varying at different frequencies in certain thickness and materials is related to tuning properties. To clarify clearly, the information about A0/S0 modes influences on relationships between amplitudes/ToFs and temperatures have been added in the last 2nd paragraph of section 4.1.
Yours Faithfully,
Tianyi

Reviewer 2 Report
The submitted manuscript with ID: “sensors-823154” and title: “Active Health Monitoring of Thick Composite Structures by Embedded and Surface Mounted Piezo Diagnostic Layer” investigates the effectiveness of embedding PZT transducers using Electro-Mechanical Impedance (EMI) method to diagnose damages in thick composite materials with regard to surface-mounted PZTs. It is an original and interesting experimental study that deals with a novel topic still open to question since the existing published work in this field of study is limited. The specific tasks of the presented research are rather clear and the paper falls within the scope of the Journal. The manuscript is easily understood, well-structured and well-presented. It also presents new results in a clear and organized manner. New concluding remarks seem to be drawn. As an overall and based on the aforementioned comments, the article could be published after revision. In order to qualify the paper for publication and to promote its objectives the Authors are encouraged to address the following comments:
- The literature background reported in Introduction is very informative. Further, in section 3 (line 194) concerning the “Sensor Integrity”) some literature review is also provided concerning the topic of PZT-based EMI method for SHM and the problem seems to be reported rather clearly. However, the important citations [39-41] are briefly presented in a few sentences (lines 195-198). Further, some recent and quite relevant studies have not been considered although they could help to point-out the gap in the existing literature that the current study is trying to fill. For example, damage diagnosis in Reinforced Concrete (RC) structures using embedded PZTs (acting as smart aggregates) and simultaneously surface bonded PZT patches, not only for the evaluation of the concrete strength gain, but also for damage detection/identification and the assessment of their severity level should be emphasized. Recent experimental studies highlighted the on-line and continuous monitoring of the potential damage evolution with time in RC structural members under shear/flexural monotonic and cyclic/seismic loading using both types of PZT transducers, as the current paper investigates. Further, recent developments of portable, real-time, wireless PZT-based impedance/admittance SHM systems which have easily and successfully applied to large-scaled RC members of existing RC structures should be reported. Furthermore, the implementation of a network of small-sized PZTs, instead of individual (single) transducers, provides more accurately results concerning the evaluation of concrete strength gain characteristics and the diagnosis of damages in RC structural members. The signal measurements of a series of surface-mounted or internally embedded PZTs arrayed in specific locations of the examined concrete member improve the damage assessment procedure providing better correlation of the local damage level and the values of the used damage index. Especially in cases of shear-critical RC members, which exhibit brittle sudden diagonal cracking, the damage index values calculated by the network of the installed PZTs lead to a safe determination of the locus and the magnitude of the occurred damage at different levels prior the fatal failure (early damage stages). Thus, based on the aforementioned comments it is strongly suggested the findings of the following additional references (ordered by date) to be considered in order to further establish the research significance and to promote the objectives of this study:
- “Damage detection in concrete structures using a simultaneously activated multi-mode PZT active sensing system: numerical modelling”, Structure and Infrastructure Engineering, 2014.
- “Combined embedded and surface-bonded piezoelectric transducers for monitoring of concrete structures”, NDT&E Int, 2014.
- “Detection of flexural damage stages for RC beams using piezoelectric sensors (PZT)”, Smart Structures and Systems, 2015.
“Damage evaluation in shear-critical reinforced concrete beam using piezoelectric transducers as smart aggregates”, Open Engineering, 2015.
- “Investigation of a new experimental method for damage assessment of RC beams failing in shear using piezoelectric transducers”, Engineering Structures, 2016.
- “Cyclic crack monitoring of a reinforced concrete column under simulated pseudo-dynamic loading using piezoceramic-based smart aggregates”, Applied Science 2016.
“Applications of smart piezoelectric materials in a wireless admittance monitoring system (WiAMS) to structures - Tests in RC elements”, Case Studies in Construction Materials, 2016.
- Research significance and subsequent impact of the presented study on the state of the practice are missing. A systematic literature review based on the previous recommendation will help into this direction.
- EMI Sensor Principles are rather briefly presented. Equations and variables require further explanations since in these equations the physical meaning of several factors should be reported.
- In line 128, the distance from the PZT transducer to each edge, which is 40 mm, is suggested to be depicted in a figure. Further, the “blue tack” could also be presented in Figure 10.
- In line 230 it is claimed: “According to the EMI, both embedded and surfaced-mounted PZT transducers have ideal bonding properties with the host structure”. Some justification concerning this issue is rather required since in line 227 a rather contradicted statement is reported: “…the embedded transducer was bonded from both sides, whilst the surface-mounted PZT transducer was only bonded from one side”. In this direction, the results reported in some of the articles suggested in the previous comment #1 conclude to similar findings and could help for the establishment of this issue.
- Some more clarifications are rather required concerning the following issues:
- In line 137 the term “ply-cutting pre-pregs” should properly be explained and illustrated in Figure 2.
- In line 266 the term “time of flight” requires further explanations.
- In line 286 the “Many factors” that affect the “above conclusions” should be clarified since the statement seems rather generic and unclear.
- In line 303 the statement: “the selection of adhesive film used to bond the PZT transducers was one of factors to affect the experimental results” should be explained more. The adhesive film affects the experimental results and more details about this could be presented.
- In line 406 the statement: “Hence, surface-mounted PZT transducers have higher attenuation than that of embedded PZT transducers” could be clarified in a new table with relative comparative results.
- Conclusions seem somewhat short and confusing. It is recommended conclusions to be re-write in order to summarize the main aspects and, mainly, the findings of this study, perhaps using bullet points.
- In Ref. 35 some data are missing.
Author Response
Dear Sir or Madam,
We are grateful for reviewer’s suggestion and the manuscript has been revised taking into consideration those comments. The revised manuscript has been attached in below file.
1. The literature background reported in Introduction is very informative. Further, in section 3 (line 194) concerning the “Sensor Integrity”) some literature review is also provided concerning the topic of PZT-based EMI method for SHM and the problem seems to be reported rather clearly. However, the important citations [39-41] are briefly presented in a few sentences (lines 195-198). Further, some recent and quite relevant studies have not been considered although they could help to point-out the gap in the existing literature that the current study is trying to fill. For example, damage diagnosis in Reinforced Concrete (RC) structures using embedded PZTs (acting as smart aggregates) and simultaneously surface bonded PZT patches, not only for the evaluation of the concrete strength gain, but also for damage detection/identification and the assessment of their severity level should be emphasized. Recent experimental studies highlighted the on-line and continuous monitoring of the potential damage evolution with time in RC structural members under shear/flexural monotonic and cyclic/seismic loading using both types of PZT transducers, as the current paper investigates. Further, recent developments of portable, real-time, wireless PZT-based impedance/admittance SHM systems which have easily and successfully applied to large-scaled RC members of existing RC structures should be reported. Furthermore, the implementation of a network of small-sized PZTs, instead of individual (single) transducers, provides more accurately results concerning the evaluation of concrete strength gain characteristics and the diagnosis of damages in RC structural members. The signal measurements of a series of surface-mounted or internally embedded PZTs arrayed in specific locations of the examined concrete member improve the damage assessment procedure providing better correlation of the local damage level and the values of the used damage index. Especially in cases of shear-critical RC members, which exhibit brittle sudden diagonal cracking, the damage index values calculated by the network of the installed PZTs lead to a safe determination of the locus and the magnitude of the occurred damage at different levels prior the fatal failure (early damage stages). Thus, based on the aforementioned comments it is strongly suggested the findings of the following additional references (ordered by date) to be considered in order to further establish the research significance and to promote the objectives of this study:
- “Damage detection in concrete structures using a simultaneously activated multi-mode PZT active sensing system: numerical modelling”, Structure and Infrastructure Engineering, 2014.
- “Combined embedded and surface-bonded piezoelectric transducers for monitoring of concrete structures”, NDT&E Int, 2014.
- “Detection of flexural damage stages for RC beams using piezoelectric sensors (PZT)”, Smart Structures and Systems, 2015.
-“Damage evaluation in shear-critical reinforced concrete beam using piezoelectric transducers as smart aggregates”, Open Engineering, 2015.
- “Investigation of a new experimental method for damage assessment of RC beams failing in shear using piezoelectric transducers”, Engineering Structures, 2016.
- “Cyclic crack monitoring of a reinforced concrete column under simulated pseudo-dynamic loading using piezoceramic-based smart aggregates”, Applied Science 2016.
-“Applications of smart piezoelectric materials in a wireless admittance monitoring system (WiAMS) to structures - Tests in RC elements”, Case Studies in Construction Materials, 2016.
- These references are inserted in the revised manuscript in EMI section.
2. Research significance and subsequent impact of the presented study on the state of the practice are missing. A systematic literature review based on the previous recommendation will help into this direction.
- Yes, the literature review of EMI part has been added in section 3.1.
3. EMI Sensor Principles are rather briefly presented. Equations and variables require further explanations since in these equations the physical meaning of several factors should be reported.
- Please see section 3.2
4. In line 128, the distance from the PZT transducer to each edge, which is 40 mm, is suggested to be depicted in a figure. Further, the “blue tack” could also be presented in Figure 10.
- Already depicted in the revised manuscript in section 5.1, and the figure of ‘blue tack’ has been added in Fig.10.
5. In line 230 it is claimed: “According to the EMI, both embedded and surfaced-mounted PZT transducers have ideal bonding properties with the host structure”. Some justification concerning this issue is rather required since in line 227 a rather contradicted statement is reported: “…the embedded transducer was bonded from both sides, whilst the surface-mounted PZT transducer was only bonded from one side”. In this direction, the results reported in some of the articles suggested in the previous comment #1 conclude to similar findings and could help for the establishment of this issue.
- The purpose of this sentence “…the embedded transducer was bonded from both sides, whilst the surface-mounted PZT transducer was only bonded from one side” is trying to explain the reason why slopes of imaginary part of admittance for embedded and surface-mounted PZT transducers are different, because the boundary conditions for embedded and surface-mounted PZT transducers are different.
- In addition, to evaluate the integrity of bonding properties between PZTs and host structure, new information by comparing the difference of two embedded PZTs and two surface-mounted PZTs. Because there is no change in the imaginary part of the admittance, so the integrity is good (In section 3.3).
5. Some more clarifications are rather required concerning the following issues:
- In line 137 the term “ply-cutting pre-pregs” should properly be explained and illustrated in Figure 2.
- Ply-cutting is a standard terminology used in composite manufacturing which refers to cutting individual plays (i.e. laminar). The is depicted in figure 2 to demonstrate the outlet for the PZT terminals.
- In line 266 the term “time of flight” requires further explanations.
- The last sentence of the 1st paragraph in the section 4.1 of the manuscript, the term ‘time of flight’ explained as ‘ …the time at peak amplitude of the first wave packet...’
- In line 286 the “Many factors” that affect the “above conclusions” should be clarified since the statement seems rather and unclear.
- Thanks for your suggestion, now the original sentence has been replaced as ‘there are five factors which affect the inverse effects of the amplitude and phase-shifts of travelling times’ (see in section 4.1 in the revised manuscript)
- In line 303 the statement: “the selection of adhesive film used to bond the PZT transducers was one of factors to affect the experimental results” should be explained more. The adhesive film affects the experimental results and more details about this could be presented.
- A paper from our research group related to these adhesive films is inserted in the last paragraph of section 4.1
- In line 406 the statement: “Hence, surface-mounted PZT transducers have higher attenuation than that of embedded PZT transducers” could be clarified in a new table with relative comparative results.
- A new figure was added to clearly and easily to see that ‘amplitude of surface-mounted PZTs have higher attenuation than that of embedded PZTs after impact’ (please see Fig. 16 in the revised manuscript)
6. Conclusions seem somewhat short and confusing. It is recommended conclusions to be re-write in order to summarize the main aspects and, mainly, the findings of this study, perhaps using bullet points.
- This part had been updated in the revised manuscript.
7. In Ref. 35 some data are missing.
- Already updated:
- [35] K. Dragan, M. Dziendzikowski, A. Kurnyta, A. Leski, J. Bienias, Structural health monitoring of composite structures with use of embedded PZT piezoelectric sensors, ECCM-16th European Conference on Composite Materials. Seville, 2014.
Yours Faithfully,
Tianyi

Round 2
Reviewer 2 Report
The paper is well revised and improved. All comments and questions raised from the previous review have been considered and properly addressed. Hence, the revised paper is suggested to be accepted for publication without further re-review.